# Immunoregulatory and Antimicrobial Activity of Bovine Neutrophil β-Defensin-5-Loaded PLGA Nanoparticles against *Mycobacterium bovis*

**DOI:** 10.3390/pharmaceutics12121172

**Published:** 2020-12-01

**Authors:** Zhengmin Liang, Yiduo Liu, Xingya Sun, Jingjun Lin, Jiao Yao, Yinjuan Song, Miaoxuan Li, Tianlong Liu, Xiangmei Zhou

**Affiliations:** 1Key Laboratory of Animal Epidemiology and Zoonosis, Ministry of Agriculture, National Animal Transmissible Spongiform Encephalopathy Laboratory, College of Veterinary Medicine, China Agricultural University, Beijing 100193, China; liangzm815@126.com (Z.L.); sy20193050809@cau.edu.cn (Y.L.); sunxingya2020@126.com (X.S.); mole-yao@hotmail.com (J.Y.); syinjuan@126.com (Y.S.); dragon726@126.com (M.L.); 2Department of Pathobiology, College of Veterinary Medicine, University of Illinois at Urbana-Champaign, Urbana, IL 61801, USA; vetjingjun@gmail.com

**Keywords:** PLGA, nanoparticles, bovine neutrophil β-defensin-5, *Mycobacterium bovis*, immunoregulation, antimicrobial activity

## Abstract

*Mycobacterium bovis* (*M. bovis*) is a member of the *Mycobacterium tuberculosis* complex imposing a high zoonotic threat to human health. The limited efficacy of BCG (Bacillus Calmette–Guérin) and upsurges of drug-resistant tuberculosis require new effective vaccination approaches and anti-TB drugs. Poly (lactic-co-glycolic acid) (PLGA) is a preferential drug delivery system candidate. In this study, we formulated PLGA nanoparticles (NPs) encapsulating the recombinant protein bovine neutrophil β-defensin-5 (B5), and investigated its role in immunomodulation and antimicrobial activity against *M. bovis* challenge. Using the classical water–oil–water solvent-evaporation method, B5-NPs were prepared, with encapsulation efficiency of 85.5% ± 2.5%. These spherical NPs were 206.6 ± 26.6 nm in diameter, with a negatively charged surface (ζ-potential −27.1 ± 1.5 mV). The encapsulated B5 protein from B5-NPs was released slowly under physiological conditions. B5 or B5-NPs efficiently enhanced the secretion of tumor necrosis factor α (TNF-α), interleukin (IL)-1β and IL-10 in J774A.1 macrophages. B5-NPs-immunized mice showed significant increases in the production of TNF-α and immunoglobulin A (IgA) in serum, and the proportion of CD4^+^ T cells in spleen compared with B5 alone. In immunoprotection studies, B5-NPs-immunized mice displayed significant reductions in pulmonary inflammatory area, bacterial burden in the lungs and spleen at 4-week after *M. bovis* challenge. In treatment studies, B5, but not B5-NPs, assisted rifampicin (RIF) with inhibition of bacterial replication in the lungs and spleen. Moreover, B5 alone also significantly reduced the bacterial load in the lungs and spleen. Altogether, our findings highlight the significance of the B5-PLGA NPs in terms of promoting the immune effect of BCG and the B5 in enhancing the therapeutic effect of RIF against *M. bovis*.

## 1. Introduction

Tuberculosis (TB) is a chronic infectious disease caused by *Mycobacterium tuberculosis* complex (MTBC) and caused more than 1.5 million deaths in 2018 [1]. *Mycobacterium bovis* (*M. bovis*) is an important member of MTBC with a genetical identity of more than 99% to *M. tuberculosis* [2]. Globally, bovine tuberculosis (bTB), caused by *M. bovis* infection, is estimated to affect more than 50 million cattle annually, costing approximately USD 3 billion [3]. More importantly, studies have shown that *M. bovis* may significantly contribute to human TB infection [4,5,6,7]. In China, only 1 out of the 245 isolates which possessed the *M. bovis* phenotype was identified as *M. bovis* [4]. Meanwhile, research addressing the epidemiology of human TB in the United States indicated that the annual percentages of tuberculosis cases attributable to *M. bovis* remained 1.3% to 1.6% in the United States during the years 2006–2013 [5]. One study showed that *M. bovis* accounts for 2.8% of all human TB cases in Africa and it is also responsible for 7.6% of human TB cases in Mexico [6]. However, another study revealed that approximately 30.2% of human TB was caused by *M. bovis* in Mexico [7]. Therefore, controlling bTB is important for reducing animal production losses and human TB cases. Vaccination can provide some protection against *M. bovis* infections [8,9,10]. However, the questionable efficacy [11] of the only available vaccine, Bacillus Calmette–Guérin (BCG), against human TB prompts an urgent need for more effective vaccination approaches.

The upsurge of drug-resistant tuberculosis requires alternatives to traditional antibiotics. Defensin, one of the most common types of cationic antimicrobial peptides (AMPs), represents an ancient highly conserved part of the innate immune system. Most defensins possess broad-spectrum antimicrobial activities as well as immunomodulatory functions. Importantly, AMPs can be used as potential treatment for TB [12]. Human β defensin-1 has been demonstrated to exhibit antimicrobial activity against both actively growing and dormant mycobacteria [13]. Interestingly, transgenic cattle expressing human β-defensin 3 have been demonstrated to have reduced susceptibility to *M. bovis* infection [14]. Several previous studies have shown that β-defensins are induced in the mucosa during chronic states of disease caused by bacteria [15,16,17]. A study has revealed that multiple mouse β-defensins can coordinate early during an infection to limit the growth of bacterial pathogen *Bordetella bronchiseptica* in the trachea [18]. Moreover, β-defensins have been shown to exhibit strong adjuvant potential for antiviral vaccine protection [19,20,21], and β-defensin-2 was found to improve the specific immune response against *M. tuberculosis* when employed as an adjuvant in the DNA vaccine construct [22]. Bovine neutrophil β-defensin-5 (B5) is a member of the β-defensins from bovine neutrophils [23]. Our previous studies demonstrated that the exogenously added B5 reduced the survival of both *M. tuberculosis* and *M. bovis* in vitro [24]. However, the antimicrobial effect of B5 against *M. bovis* in vivo was unknown.

Protein- or peptide-loaded nanoparticles (NPs) have been employed as efficient and stable vaccine-delivery vehicles against infectious diseases [25]. Growing studies have sought to find safe and effective vaccine adjuvants and drug delivery systems to formulate better mucosal vaccines, based on polymeric NPs [26]. One of the most commonly used polymeric NPs for vaccine delivery is Poly (lactic-co-glycolic acid) (PLGA). The favorable characteristics of using PLGA as an ideal delivery carrier include safety, biocompatibility, antigen stabilization, enhancement of antigen immunogenicity, and so on [27,28]. For example, one study has shown that single-dose Ag85B-ESAT6-loaded PLGA NPs provided long-term protective immunity against *M. tuberculosis* in mice [29]. In addition, PLGA NPs also promote the immunogenicity of vaccine adjuvant. Surface assembly on PEGylated PLGA microspheres promise to improve both efficacy and safety of poly(I:C) [30]. Furthermore, nanoparticles based on defensin have been demonstrated to efficiently protect laboratory animals against pathogens such as methicillin-resistant *Staphylococcus aereus* [31], multidrug resistant *Escherichia coli* [32], and *M. tuberculosis* [33]. However, it is unknown whether PLGA NPs can promote the immune regulation or antibacterial activity of B5. In the current study, we investigated the immunoregulatory and antimicrobial effect of B5 and its nanoparticles (B5-NPs) against *M. bovis* in mice.

## 2. Methods

### 2.1. Mice

Specific-pathogen-free 6–8-week-old female BALB/c mice were purchased from SPF Biotechnology (Beijing, China). The mice were kept in the biosafety level 3 (BSL3) facilities, College of Veterinary Medicine, China Agricultural University, Beijing. The mice were housed in a special pathogen-free room with food and water ad libitum. All the animal experiments were carried out in accordance with the Chinese Regulations of Laboratory Animals—The Guidelines for the Care of Laboratory Animals (Ministry of Science and Technology of People’s Republic of China) and Laboratory Animal Requirements of Environment and Housing Facilities (GB 14925–2010, National Laboratory Animal Standardization Technical Committee). The animal studies and research protocols were approved by The Laboratory Animal Ethical Committee of China Agricultural University and the license number of the research protocol was cau20170108-2 (8 January 2017).

### 2.2. Bacterial Culture

BCG and *M. bovis* (*M. bovis* C68004) were grown in 7H9 broth (Difco, New York, NY, USA) supplemented with 2 g/L sodium pyruvate, 0.05% Tween 80, and 10% OADC (Oleic acid, Albumin, Dextrose, Catalase) enrichment solution (BD Biosciences, New York, NY, USA) under constant shaking at 37 °C. BCG was aliquoted, frozen at approximately 3 × 10^8^ CFU/mL in single-use aliquots, and stored at −80 °C. Immediately before injection, BCG was thawed and diluted in cold PBS. The *M. bovis* culture was cultured to the log phase (OD 0.4–0.9) when sterile glycerol (20%) was added, and single-use aliquots at approximately 3 × 10^8^ CFU/mL were frozen and stored at −80 °C until needed. Single cell suspensions were obtained by passing the bacteria 10–15 times through a 22 Gauge needle [34,35].

### 2.3. Preparation of B5-NPs

Recombinant B5 protein was prepared as previously described [24]. Briefly, the recombinant yeast strains pPIC9K-B5-His-GS115 expressing B5 were cultivated at 28 °C in Buffered Glycerol-complex (BMGY) medium for 24 h, then the cells were incubated at 28 °C for 72 h in Buffered Methanol-complex (BMMY) medium containing 0.5% methanol (methanol was added every 24 h). The culture supernatant was collected and purified with Ni^2+^ Sepharose High Performance (GE Healthcare Life Sciences, Piscataway, NJ, USA). Protein identity was confirmed by Tricine-SDS (sodium dodecyl sulfate)-PAGE (polyacrylamide gel electrophoresis) followed by Coomassie blue staining and immunoblotting with mouse monoclonal antibody against His·tag (1:5000). PLGA polymer (the ratio of lactide:glycolide feed was 50:50, Mw: 10KDa, Jinnan Daigang Biomaterial Co., Ltd., Shandong, China) was used for B5 encapsulation. PLGA NPs were generated utilizing a water–oil–water emulsion technique as reported [29,36]. B5 was encapsulated in PLGA particles, followed by solvent evaporation, washing, and drying. Briefly, 1 mg B5 protein in 500 µL of PBS (aqueous phase) was emulsified with 4.5 mL of ethyl acetate dissolving 160 mg PLGA (organic phase). Emulsification was performed in an ice bath by sonication at 60 W for 6 min using a 2 mm stepped microtip in a 950 W ultrasonic processor (Scientz, Zhenjiang, China). For preparing blank NPs (PBS-NPs), PBS was used as the internal aqueous phase. The obtained water–oil emulsion was immediately added dropwise to 10 mL of 1% polyvinyl alcohol (PVA) solution (aqueous phase). The water–oil–water emulsion was then resonicated with a 2 mm stepped microtip at 60 W for 8 min. Subsequently, the emulsion was immediately added dropwise to 10 mL of 0.5% PVA solution and immediately stirred for 4 h at room temperature for ethyl acetate evaporation and hardening of NPs. Finally, the B5-PLGA NPs were harvested by centrifugation and washing with ice-cold distilled water three times. The size and zeta potential of NPs were measured by a Zetasizer Nano ZS system (Malvern Instruments Ltd., Worcestershire, UK). Surface morphology of the NPs was examined using scanning electron microscopy (EVO40; Carl Zeiss, Jena, Germany).

### 2.4. Determination of B5-Encapsulation and -Loading Efficiency

The encapsulation efficiency (EE) and loading efficiency (LE) were measured using micro-BCA protein assay kit (Thermo Fisher Scientific, Carlsbad, CA, USA) [29]. Briefly, 10 mg NPs were dissolved in 1 mL acetonitrile (Sigma-Aldrich, Louis, MO, USA), followed by centrifugation at 13,000 g for 10 min. The procedure was repeated thrice to remove PLGA, and the B5 protein pellet obtained was solubilized in 50 μL of SDS (1%). B5 EE was estimated by micro-BCA using BSA standards. B5 protein integrity was monitored by SDS-PAGE upon Coomassie brilliant blue staining using purified B5 as control. B5-NPs EE and LE were calculated:(1)$$\mathrm{EE}\%=\frac{\mathrm{Weight}\;\mathrm{of}\;B5\;\mathrm{encapsulated}}{\mathrm{Total}\;\mathrm{weight}\;\mathrm{of}\;B5\;\mathrm{used}\;\mathrm{for}\;\mathrm{encapsulation}}\;\times \;100\%$$
(2)$$\mathrm{LE}\%=\frac{\mathrm{Weight}\;\mathrm{of}\;B5\;\mathrm{encapsulated}}{\mathrm{Total}\;\mathrm{dry}\;\mathrm{weight}\;\mathrm{of}\;\mathrm{NPs}}\;\times \;100\%$$

### 2.5. In Vitro Release Kinetics of B5-NPs

The in vitro release kinetics of B5-NPs were measured using a micro-BCA protein assay kit. To monitor the in vitro release profile of B5-NPs under physiological conditions, 50 mg B5-NPs were suspended in 1 mL of PBS (pH 7.4) and incubated at 37 °C with constant stirring at 80 rpm. Supernatants were collected at regular intervals (until 7 days) and the release of B5 protein was estimated by micro-BCA protein assay kit. The B5 protein released in the medium was calculated:(3)$$\mathrm{Protein}\;\mathrm{release}\;(\%)=\frac{B5\;\mathrm{released}}{B5\;\mathrm{encapsulated}}\;\times \;100\%$$

### 2.6. Cell Culture and Stimulation

J774A.1 macrophages were obtained from the Cell Culture Center, Peking Union Medical College (Beijing, China) and cultured in DMEM containing fatal bovine serum (10%), streptomycin (100 μg/mL) and penicillin (100 U/mL) at 37 °C with 5% CO_2_ in humidified incubator. Then, the macrophages were moved to 12-well cell culture plates (5 × 10^5^ cells in each well) for 12 h prior to stimulation. The cells were stimulated respectively with B5 (5 μg/mL), B5-NPs (100 μg/mL) or PBS-NPs (100 μg/mL) for 24 h. Untreated cells were taken as negative controls. The cell culture supernatants were collected to quantify the levels of cytokines.

### 2.7. Mice Immunization and M. bovis Challenge

To evaluate the immunoregulatory activity of B5 and B5-NPs, mice were vaccinated subcutaneously with 10^6^ CFU of vaccine strains in 100 μL of PBS (Figure 1A). Starting at week 4, each mouse was intranasally immunized with totally 3 doses of B5 (50 µg in 50 µL of PBS, equivalent to 2.5 mg/kg) or B5-NPs (1 mg B5-NPs in 50 μL of PBS, equivalent to 50 mg/kg) three weeks apart. Control mice received intranasal injections of PBS-NPs or PBS. Immunized mice were intranasally challenged with ~1000 CFU of *M. bovis* in 50 µL of PBS three weeks after the last immunization. A group of infected mice were killed 1 day after challenge to determine the initial bacterial load in lungs, which resulted to be ~950 CFU. The mice were euthanized four weeks after challenge.

To evaluate the antimicrobial activity of B5 and B5-NPs, mice were intranasally challenged with ∼100 CFU of *M. bovis* in 50 µL of PBS (Figure 1B). Four weeks later, *M. bovis*-infected mice were subcutaneously treated with B5 (5 mg/kg) or B5-NPs (100 mg/kg) three times a week for two weeks. Both the B5 and B5-NPs group were divided into two subgroups, one with daily rifampicin (RIF, 10 mg/kg) treatment and the other one without for the same two-week course. A group of infected mice were killed 1 day after challenge to determine the initial bacterial load in the lungs, which resulted to be ∼110 CFU. The mice were euthanized two weeks after the last treatment.

### 2.8. Enzyme-Linked Immunosorbent Assay (ELISA) Assay

The levels of TNF-α, IL-1β, IL-10 and IgA were measured by ELISA kits (Neobioscience, Shenzhen, Guangdong, China) according to the manufacturer’s instructions. Briefly, standards or samples (100 µL) were added to 96-well ELISA plates and incubated at 37 °C for 90 min. The supernatants were discarded, and the ELISA plates were washed with a washing buffer 4 times. The ELISA plates were then incubated by detection antibody solutions (100 µL), HRP-conjugated antibodies (100 µL), and TMB substrates (100 µL). After adding the stop solution (100 µL), the optical density (OD) was obtained at 450 nm by using an ELISA Plate Reader (Thermo Scientific Multiskan FC, Shanghai, China). A standard curve was obtained using two-fold dilutions of the standard for each independent experiment. Samples were added in triplicates in each independent experiment.

### 2.9. Flow Cytometry

Single-cell suspensions of splenocytes were washed twice with cell staining buffer from Multi Sciences LTD (Zhejiang, China) and stained with antibodies for PerCP-Cy5.5-conjugated anti-CD3, FITC-conjugated anti-CD4, APC-conjugated anti-CD8 from Multi Sciences LTD (Zhejiang, China) at 4 °C for 30 min. Then the cells were washed twice with cell staining buffer. The number of each type of T cell was measured by a FACSVerse flow cytometer (BD Biosciences) and analyzed with FlowJo software (TreeStar, Ashland, OR, USA).

### 2.10. Colony-Forming Units (CFU) Enumeration

For CFU assay in tissues, tissues of lung and spleen from sacrificed mice were homogenized in 1 mL 7H9 broth (BD) supplemented with 2 g/L sodium pyruvate, 0.05% Tween 80, and 10% OADC using OmniTip Plastic Homogenizer Probes (Omni International, Bedford, MA, USA). Serial dilutions in PBS + 0.05% Tween 80 were plated on 7H10 agar plates supplemented with OADC (10%), amphotericin B (10 µg/mL) and polymyxin B sulfate (10 µg/mL). Plates were then incubated at 37 °C and the number of CFU was counted after 3 weeks.

### 2.11. Lung Histology

A single lobe of the lung was fixed in 10% normal buffered formalin. Tissues were embedded in paraffin, slides were prepared, and stained with Hematoxylin and eosin (H&E) or acid-fast bacilli (AFB) after fixation. Between 3 and 16 nonoverlapping fields per tissue section were captured at a magnification of ×100 to ×1000. Qualitative analyses were carried out, comparing all fields within a group (four animals per group, two to three sections examined per animal) for inflammatory areas. Inflammatory areas were quantitatively analyzed under low magnification with Image-Pro plus 6.0 software (National Institutes of Health, Bethesda, MA, USA) [37].

### 2.12. Statistical Analysis

All data are expressed as the mean ± standard deviation (SD). For statistical analysis, one-way analysis of variance (ANOVA) followed by the Dunnett’s multiple comparison test was performed, and graphs were generated using GraphPad Prism version 7.0 (GraphPad Software Inc., San Diego, CA, USA). Results of the flow cytometry were analyzed by the FlowJo software version 10 (Treestar, CA, USA). A *p*-value less than 0.05 reflected that the findings were statistically significant. A *p*-value less than 0.05 reflected the findings statistically significant.

## 3. Results

### 3.1. Physicochemical and Morphological Characterization of B5-NPs

Soluble B5 was purified by Ni^2+^ Sepharose High Performance. The purified B5 was analyzed using Tricine-SDS-PAGE and Western blotting. There was the sole band (>95% purity) at the expected molecular mass of 7 kDa on Tricine-SDS-PAGE after Coomassie brilliant blue staining (Figure 2A) and on the PVDF membrane probed using anti-His·tag monoclonal antibody (Figure 2B). B5-NPs were prepared by double-emulsion solvent evaporation. We obtained B5 protein EE and LE of 85.5% ± 2.5% and 0.5% ± 0.02% (Table 1), respectively. The rate of B5 release from B5-NPs was estimated with the micro-BCA kit at regular intervals until 7 days under in vitro conditions. Surprisingly, the in vitro release profile of B5 protein from B5-NPs showed that after only 2 h of incubation, a burst release of 18.25% ± 2.2% of B5 was detected in the supernatant (Figure 2C). By 12 h, 37.25% ± 2.2% of B5 had been released in the supernatant. However, after 24 h, the release profile had reached a plateau and thereupon displayed controlled release. On day 7, slow and sustained release of 67.5% ± 2.4% of the total B5 protein was observed, suggesting that in vivo administrated B5-NPs should also be able to slowly release B5 over a longer duration to stimulate the immune system. Scanning electron microscopy of B5-loaded PLGA NPs revealed a uniform, smooth appearance, globular, with no cavities (Figure 2D,E). Dynamic light scattering of B5-NPs suspension on the Zetasizer Nano ZS indicated narrow size distribution and greater colloidal stability. Mean particle diameter was found to be 206.6 ± 26.6 nm in PBS (pH 7.4). The ζ-potential of these particles was found to be −27.1 ± 1.5 mV (Table 1, Figure 2F,G).

### 3.2. B5-NPs Enhance the Cytokine Secretion in J774A.1 Cells

Inflammatory cytokines are important for the activation and recruitment of immune cells to sites of bacterial infection. We therefore examined whether B5 and B5-NPs influence the expression of the pro-inflammatory cytokines TNF-α and IL-1β, and the anti-inflammatory cytokine IL-10 in J774A.1 macrophages. B5-NPs stimulated cells exhibited markedly higher expression of TNF-α and IL-10 than those stimulated with B5 or PBS-NPs after 24 h (Figure 3A,C). Meanwhile, there were no difference in IL-1β production between B5 and B5-NP groups (Figure 3B). Together, these data indicate that B5 and B5-NPs induce both pro-inflammatory and anti-inflammatory cytokine response in vitro.

### 3.3. B5-NPs Promote TNF-α and IgA Production in Mice

To investigate the immunoregulatory activity of B5 and B5-NPs in vivo, cytokine and antibody levels in serum or bronchoalveolar lavage fluid (BALF) were analyzed three weeks after the last immunization, but before infection with live *M. bovis*. Immunization with B5 significantly increased the levels of IL-1β and IL-10, but not TNF-α in serum (Figure 4A–C). B5-NPs exhibited higher levels of TNF-α and IL-10 compared with B5, while both B5 and B5-NPs resulted in significant increase in IgA production (Figure 4E,F). However, neither B5 nor B5-NPs could increase IgG production (Figure 4C,D). As control, PBS-NPs treatment did not show any significant difference in our cytokine assays. Above all, immunization with B5-NPs resulted in higher levels of TNF-α and IgA than B5, as expected (Figure 4).

### 3.4. B5-NPs Increase the Population of CD4^+^ and CD8^+^ T Cells in Spleen

A flow cytometric analysis of the CD3^+^ T cells in spleen showed that there was a significant increase in the population of CD3^+^ CD4^+^ and CD3^+^ CD8^+^ T cell subsets in BCG-immunized mice treated with B5-NPs, compared with the BCG or BCG + PBS-NP group (Figure 5). B5-NPs also induced significantly higher CD3^+^ CD4^+^ T cell percentages than the B5 group (Figure 5A,B). There was no significant difference in these cells between BCG group and BCG + PBS-NP group.

### 3.5. B5-NPs Enhance Protective Efficacy against M. bovis

B5-NPs efficacy was subsequently tested in mice challenged with *M. bovis* strains. Animals were immunized with B5, B5-NPs, PBS-NPs or PBS (normal control) and challenged intranasally with *M. bovis*. The size of inflammatory area was significantly different between BCG group and BCG + B5 or BCG + B5-NP group, as shown on lung tissues by H&E staining (Figure 6A,G). Furthermore, numerous acid-fast bacilli (AFBs) were observed throughout the lung sections of all immunized mice, whereas fewer AFBs were found in the lungs of mice in BCG+ B5-NP group compared with BCG group. AFB results correlated well with the CFU of bacteria observed in these animals (Figure 6B,E). CFU enumeration results indicated that BCG combined with B5-NPs led to significant reductions in lung bacilli and spleen bacilli compared to the BCG control. Immunization with B5 also showed lower numbers of bacteria in the lungs and spleen, however the difference was statistical insignificant (Figure 6E,F). Moreover, in comparison to mice in the PBS group, the relative weight (organ coefficient, indicating severity of inflammation and edema) of lung in *M. bovis*-infected mice was increased, and BCG combined with B5-NPs remarkably decreased the organ coefficient of lung compared with BCG control (Figure 6C). However, there were no significant differences in the organ coefficient of spleen among immunized groups (Figure 6D). In addition, few differences in pulmonary inflammatory area and bacterial burden in the lungs or spleen between BCG group and BCG + PBS-NP group were observed.

### 3.6. B5 Promotes the Therapeutic Effect of RIF against M. bovis

The observed protective efficacy of B5-NPs and antimicrobial effect of B5 against *M. bovis* in vitro [24] encouraged us to test the ability of B5-NPs to control *M. bovis* infection. As a model for post-exposure strategy (Figure 1B), BALB/C mice were infected via intranasal exposure and then treated with B5 or B5-NPs. Histological examination showed that severe inflammation and plenty of bacteria were visible in the *M. bovis* group compared with PBS group (normal control), and all other groups, including RIF, RIF + B5, RIF + B5-NPs, B5, and B5-NPs, showed significantly alleviated pathological damage of lungs, and reduced pulmonary inflammatory area and bacterial number. Notably, the inflammatory area and the dissemination of *M. bovis* bacilli in lung tissues in RIF + B5 group were significantly reduced compared with RIF groups (Figure 7A,B,G). RIF + B5 treatment also significantly reduced the number of viable bacteria in the lungs and spleen. Unexpectedly, few differences in bacteria counts of the lungs or spleen were seen between RIF group and RIF + B5-NP group (Figure 7E,F). Additionally, there was no significant difference in the organ coefficient of lung or spleen among RIF-treated groups (Figure 7C,D). It was very interesting to note that the number of bacteria in the lungs and spleen of B5 group was significantly decreased compared with that of *M. bovis* group (Figure 7E,F). Furthermore, some differences in bacterial burden in the lungs and spleen between B5-NP group and *M. bovis* group were noticed, but they did not achieve statistical significance. These results suggested that B5 could help RIF inhibit bacterial replication in vivo.

### 3.7. B5 and B5-NPs Reduce M. bovis-Induced TNF-α Hypersecretion

BALF is an important technique to examine inflammation, immune, cytotoxic responses, and disease advancement in respiratory airways. The pro-inflammatory cytokine TNF-α is a critical immune mediator in the protection against and pathology of tuberculosis [38]. The level of TNF-α was analyzed as an indicator of inflammation in the lung airways after treatment, and evaluation of TNF-α may provide additional insight to the relationship between disease pathology and therapeutic effect. We found that treatment with RIF, or combined with B5 or B5-NPs, respectively, evidently decreased the *M. bovis*-induced high level of TNF-α in serum and BALF. Meanwhile, RIF combined with B5 strikingly reduced the concentration of TNF-α in BALF compared with the RIF control. Interestingly, B5-NPs alone significantly decreased TNF-α secretion in serum and BALF (Figure 8).

## 4. Discussion

The upsurge of drug-resistant tuberculosis requires alternatives to traditional anti-TB drugs. AMPs are efficient candidates to fight against tuberculosis [12]. Not surprisingly, much effort has been made to develop AMPs as anti-infective therapeutics for clinical use in the past decades [39]. Despite the efforts, little success is achieved because of the inherent pharmacological limitations of small peptides [40]. However, the discovery of anti-mycobacterium activities of β-defensin 2 along with advances in nanobiotics [20,21,41], strongly hints that combining host-defense peptide with nanobiotics can improve its pharmacological properties and thereby display better anti-microbial activities even against the drug-resistant bacteria. PLGA NPs-based delivery systems have emerged as promising next-generation vaccination strategies. In the current study, we investigated the immunoregulatory and antimicrobial activity of B5 and its PLGA-based nanoparticles B5-NPs against *M. bovis*.

Typically, PLGA NPs have negative ζ-potential due to their carboxyl groups [42]. NPs of 200–300 nm have been showed to induce a strong cellular immune response as well as a heightened humoral response [43]. Similarly, morphological characterization of B5 PLGA NPs showed spherical NPs with an average diameter of 206.6 ± 26.6 nm and zeta potential of −27.1 ± 1.5 mV. Earlier studies have highlighted the role of PLGA in protein protection from proteolysis in vitro or in vivo [44,45]. In agreement with these typically observed NP values, the B5 protein released by B5 PLGA NPs was shown to be intact, with no signs of degradation for at least seven days under physiological conditions.

TNF-α is essential for the control of mycobacterial infections [38]. IL-1β is important for the development of antimicrobial adaptive immunity during the early stages of mycobacterial infection [46]. IL-10 has been reported to promote mycobacterial persistence [47] but also may have protective effects in TB by limiting pathologic inflammatory responses [48,49]. B5-loaded PLGA NPs significantly upregulated the secretion of TNF-α, IL-1β and IL-10 in macrophages, similar to the data previously reported for other adjuvant targets such as Alhagi honey polysaccharide [50] and TLR ligands [51]. To expand upon our in vitro studies, we assessed the efficacy of B5-NPs, relative to B5, to prevent lung damage in a mouse model of *M. bovis* infection. In this current study, a new immunization strategy was tested and achieved greater protection against *M. bovis*, based on the use of B5-loaded PLGA NPs to reduce the number of bacteria in lungs and spleen and alleviate the pathological damage of lung. To assess the immunoregulatory activity of B5-NPs, we analyzed levels of cytokines and antibodies in immunized mice. In line with experimental results in vitro, B5-NPs induced much higher levels of TNF-α than B5 in mice. Furthermore, there was a significant increase in the proportion of CD3^+^ CD4^+^ and CD3^+^ CD8^+^ T cell subsets in the B5-NPs mouse group. CD4^+^ and CD8^+^ T-cell immunity is critical for controlling TB [52]. Thus, it indicated that B5-NPs may act as a new immunomodulator to promote the proliferation of CD4^+^ and CD8^+^ T cells for the control of tuberculosis. In addition to T-cell immunity, a growing number of studies provide evidence on antibody-mediated immunity against *M. Tuberculosis* [53,54]. IgA plays an important role in mucosal immunity against TB; a study of a rhesus macaque model has shown that a high level of IgA in BALF was associated with protection against *M. tuberculosis* infection [55]. Consistent with the previous studies, our results revealed IgA secretion in both serum and BALF was significantly induced by B5, and the PLGA nanoparticles enhanced this effect. Together, these results indicate the utility of B5-NPs loading a low dose of B5 can provide and enhance protection in a murine model of bTB, relative to immunization with a high dose of B5.

To date, several studies have shown that β-defensin regulates not only innate immune but also adaptive immune responses against virus or *M. tuberculosis* [19,20,21,22], these studies have focused on adjuvant activity of β-defensin and combination with pathogen antigen as a subunit vaccine. Although studies have demonstrated some success using various nanoparticles in promoting antibacterial activity and wound healing of AMPs [31,32,56], the effect of AMP nanoparticles on the immune response against the intracellular bacteria *M. bovis* is unclear. Effective adjuvants are categorized based on two composition, delivery systems, and their ability to trigger innate immune activation [57]. Adjuvants which can be administered safely via mucosal routes, may have the biggest impact on future directions in TB vaccine design [58]. In this work, the ability of B5 encapsulated in PLGA NPs to improve innate responses and confer protection against *M. bovis* indicated that B5 and its nanoparticles B5-NPs exhibit a strong adjuvant potential of mucosal immunity.

Several AMPs from humans and mice have been shown to exhibit potential for TB treatment [12,13]. However, the antibacterial activity of AMPs from the naturally infected animal of tuberculosis against *M. bovis* is unknown. In this study, we investigated the antimicrobial effect of B5 from the bovine neutrophils in mice. Our results showed that B5 increased the therapeutic effect of RIF against *M. bovis* by reducing bacterial load and inflammatory area in lungs. In addition, we have also demonstrated that the bacterial load of lung tissues in 10 mg/kg RIF with B5 treatment group is lower than that of the 20 mg/kg RIF alone treatment group, suggesting that low-dose RIF combined with B5 treatment could achieve an ideal antibacterial effect (data not shown). Furthermore, our results also indicated that B5 alone significantly inhibited bacterial growth in mice, which was consistent with our previous observations of experiments in vitro [24]. It is worth mentioning that the generic, broad-spectrum activity of AMPs has faced challenges by issues such as specificity and drug resistance [39]. To resolve this problem, AMPs from animals and plants can be used to substitute human AMPs, which may limit the risk of cross-resistance to endogenous host defense [39]. Therefore, it is possible to use B5 in human tuberculosis treatment, but further study on its antibacterial mechanism, safety and dosage will be needed.

It has been shown that PLGA-based NPs improve the efficacy of the antimicrobial peptide plectasin against Staphylococcus aureus infections in epithelial tissues [41], and peptide-modified PLGA NPs provide more efficacious protection against Porphyromonas gingivalis-induced inflammation in vivo [59]. Additionally, a study has reported the enhanced therapeutic potential of host defense peptide IDR-1018 against *M. tuberculosis* via N-acetyl cysteine decorated porous PLGA microspheres [33]. However, in the current study, PLGA NPs alone failed to promote the antibacterial activity of B5. Although B5-NPs could significantly reduce the high level of TNF-α caused by *M. bovis*, it could not enhance clearance of *M. bovis* by RIF. This is very likely caused by the relatively lower amount of B5 released from B5-NPs than freely diffused B5. In conclusion, B5 could be used as immune adjuvant to improve the immune effect of BCG and its nanoparticles enhance the effect, moreover, B5 could also serve as an adjunct chemotherapy molecule to promote the therapeutic effect of rifampicin against tuberculosis.

## Figures and Tables

**Figure 1 pharmaceutics-12-01172-f001:**
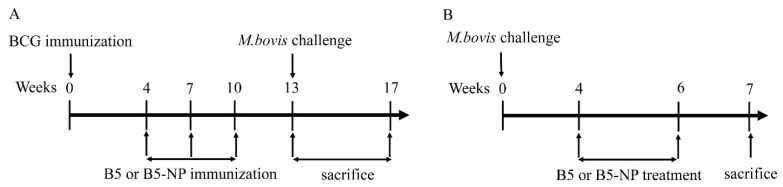
Experimental design for evaluation of immunoregulatory and antimicrobial activities of B5 protein nanoparticles (B5-NPs). (**A**) Flow chart depicting the pre-exposure strategy. Groups of BALB/c mice (n = 10 per group) were immunized subcutaneously with 10^6^ CFU of Bacillus Calmette–Guérin (BCG) in 100 µL of PBS. At four weeks before infection, mice were immunized intranasally with B5 (2.5 mg/kg) or B5-NPs (50 mg/kg) three times every three weeks. (**B**) Flow chart depicting the post-exposure strategy. Mice were challenged intranasally with 110 CFU of *M. bovis*. After 4 weeks, the mice were treated subcutaneously with B5 (5 mg/kg), B5-NPs (100 mg/kg), or respectively combined with rifampicin (RIF) (10 mg/kg) until week 6.

**Figure 2 pharmaceutics-12-01172-f002:**
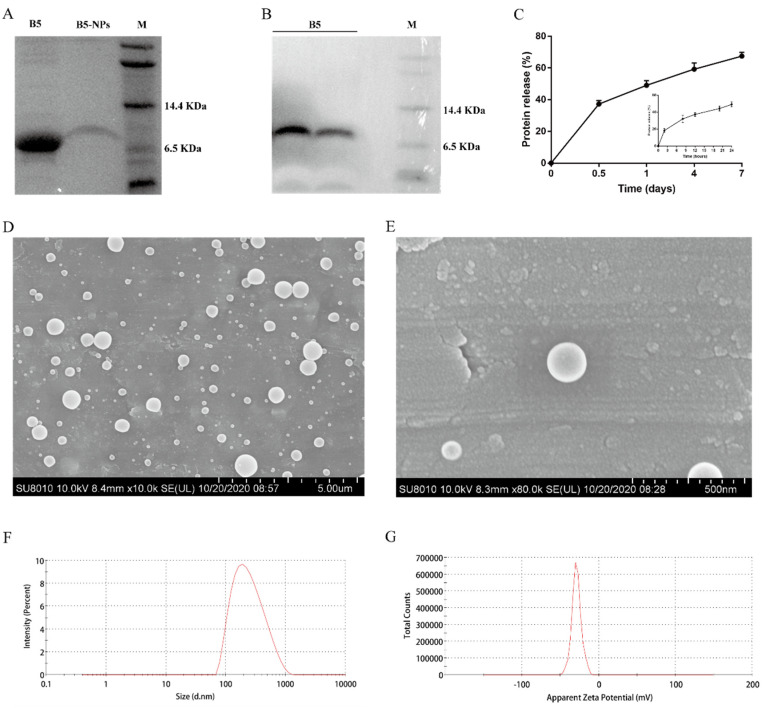
Physicochemical and morphological characterization of B5-NPs. (**A**) The purified recombinant B5 from B5-expressing *pichia pastoris* cells that was used for the preparation of B5-NPs (left lane). The protein recovered from B5-NPs after 7 days of in vitro release (middle lane) and protein molecular weight ladder (right lane) were subjected to Tricine-SDS-PAGE followed by Coomassie blue staining. (**B**) Western blotting identification of the purified B5 by using anti-His·tag antibody. (**C**) The release profile of B5 protein from B5-NPs in vitro. B5-NPs suspended in PBS (50 mg/mL) were incubated at 37 °C for indicated time periods, and the release of B5 protein was estimated by micro-BCA assay. The initial burst release caused >45% of B5 protein to be released within 24 h (inner graph), followed by slower-release kinetics that reached ∼68% of B5 release by 7 days (outer graph). (**D**,**E**) Scanning electron microscopy observed the morphological characterization of B5-NPs. (**F**,**G**) The physicochemical characterization of B5-NPs was determined by dynamic light scattering (**F**) and ζ-potential analysis (**G**).

**Figure 3 pharmaceutics-12-01172-f003:**
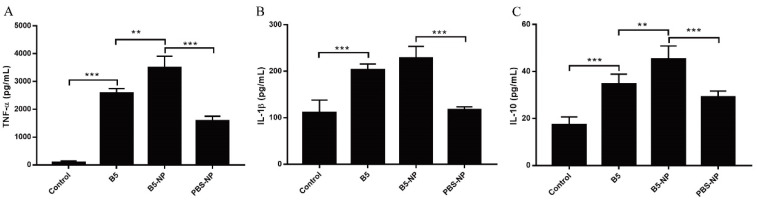
B5 and B5-NPs induce pro-inflammatory and anti-inflammatory cytokine production in macrophages. (**A**–**C**) ELISA analysis of TNF-α (**A**), IL-1β (**B**), and IL-10 (**C**) production in J774A.1 macrophages stimulated with B5, B5-NPs, or PBS-NPs for 24 h. B5: BNBD5, B5-NP: BNBD5- PLGA NPs, PBS-NP: PBS- PLGA NPs. All data are shown in vitro representing the mean ± SD of three independent experiments (** *p* < 0.01; *** *p* < 0.001).

**Figure 4 pharmaceutics-12-01172-f004:**
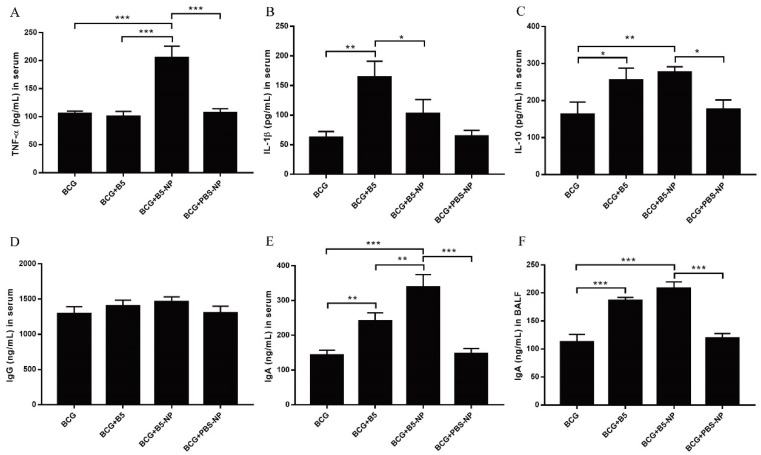
B5-NPs induce cytokines and antibodies production in BCG-immunized mice. (**A**–**E**) Concentrations of TNF-α (**A**), IL-1β (**B**) and IL-10 (**C**) in serum, IgG (**D**) in serum and IgA in serum (**E**) or bronchoalveolar lavage fluid (BALF) (**F**) were examined by ELISA. Serum and BALF were collected three weeks after the last immunization. All data shown represent the mean ± SD of three independent experiments (* *p* < 0.05; ** *p* < 0.01; *** *p* < 0.001).

**Figure 5 pharmaceutics-12-01172-f005:**
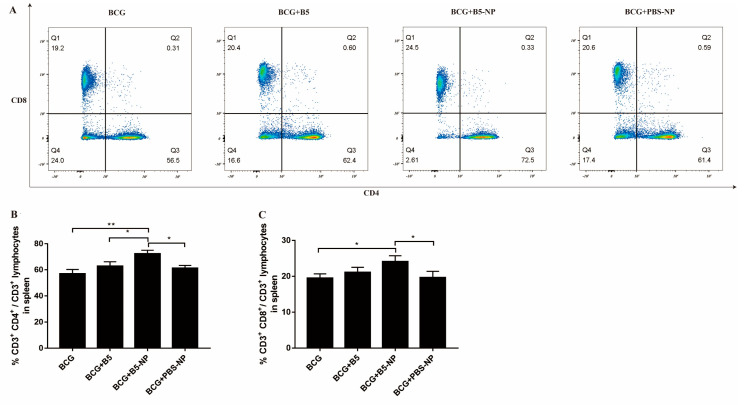
The percentage of CD3^+^ CD4^+^ and CD3^+^ CD8^+^ lymphocytes within splenocytes. (**A**–**C**) Flow cytometry analysis percentage of CD3^+^ CD4^+^ and CD3^+^ CD8^+^ lymphocytes within splenocytes. At week 13, isolated spleen cells from mice were stained with PerCP-Cy5.5-conjugated anti-CD3, FITC-conjugated anti-CD4, APC-conjugated anti-CD8. CD3^+^ CD4^+^ and CD3^+^ CD8^+^ T-cell subgroup were detected via flow cytometry. Data collected are expressed as the mean ± SD of three independent experiments (* *p* < 0.05; ** *p* < 0.01).

**Figure 6 pharmaceutics-12-01172-f006:**
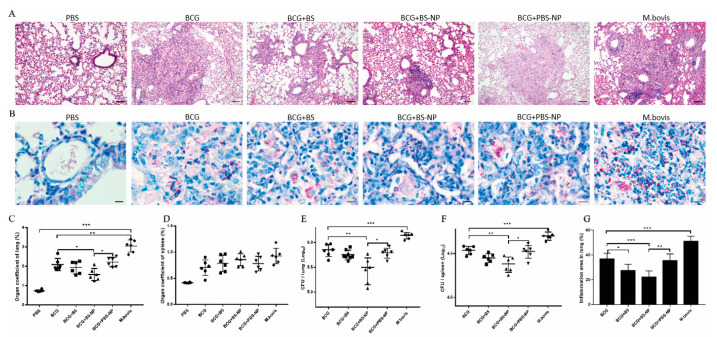
B5-NPs reduce pathological damage of lung. (**A**,**B**) Histopathological evaluation of lung tissues was performed with hematoxylin and eosin (H&E) staining (**A**) and acid-fast staining (**B**) four weeks after challenge with *M. bovis* strain. The top panel (**A**) shows images at ×100 magnification (scale bars, 100 μm); the bottom panel (**B**) shows images at ×1000 magnification (scale bars, 10 μm). (**C**–**F**) The organ coefficient of lung (**C**) and spleen (**D**), the number of viable bacteria in the lungs (**E**) and spleen (**F**) were determined 4 weeks after challenge with *M. bovis* strain. (**G**) Percentage of lung area involved in inflammation lesions relative to total lung area used for morphometric analysis. Data shown are representative of six mice per group. Data collected are expressed as the mean ± SD. (* *p* < 0.05; ** *p* < 0.01; *** *p* < 0.001).

**Figure 7 pharmaceutics-12-01172-f007:**
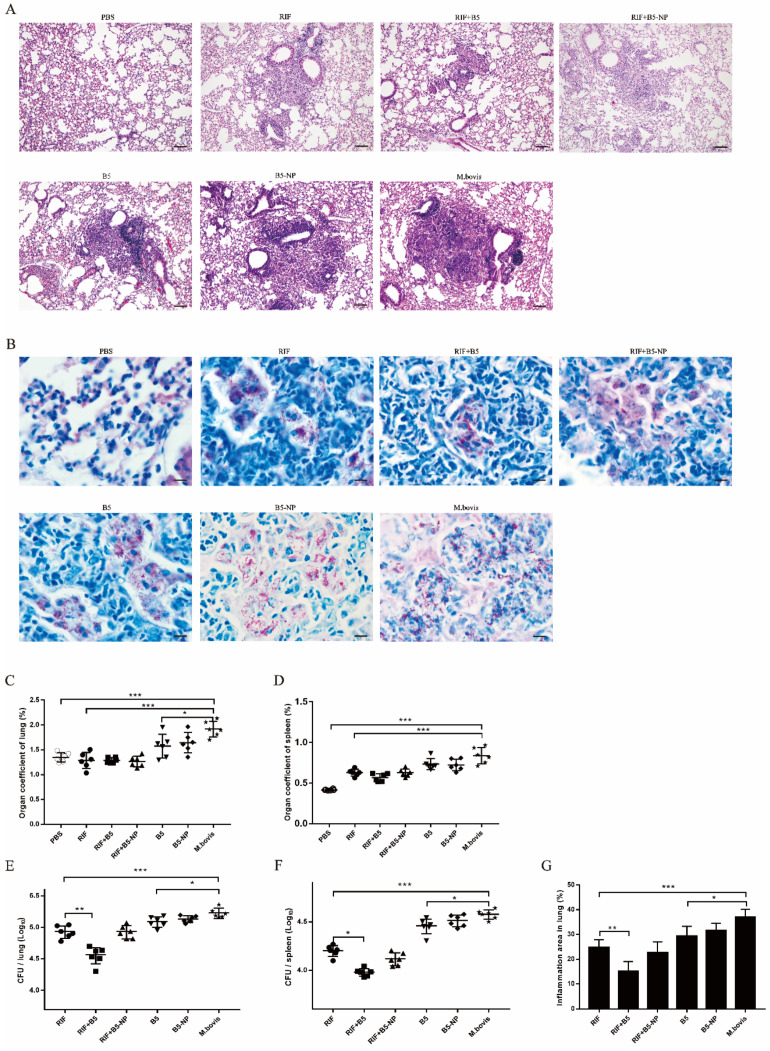
The therapeutic effect of B5 combined with RIF on *M. bovis*-infected mice. (**A**,**B**) Histopathological evaluation of lung tissues of *M. bovis*-infected mice was performed with H&E staining (**A**) and acid-fast staining (**B**) two weeks after treatment with B5, B5-NPs, or combined with rifampicin (RIF), respectively. The top panel (**A**) shows images at ×100 magnification (scale bars, 100 μm); the bottom panel (**B**) shows images at ×1000 magnification (scale bars, 10 μm). (**C**–**F**) The organ coefficient of lung (**C**) and spleen (**D**), and the number of viable bacteria in the lungs (**E**) and spleen (**F**) were determined 2 weeks after treatment. (**G**) Percentage of lung area involved in inflammation lesions relative to total lung area used for morphometric analysis. Data shown are representative of six mice per group. Data collected are expressed as the mean ± SD. (* *p* < 0.05; ** *p* < 0.01; *** *p* < 0.001).

**Figure 8 pharmaceutics-12-01172-f008:**
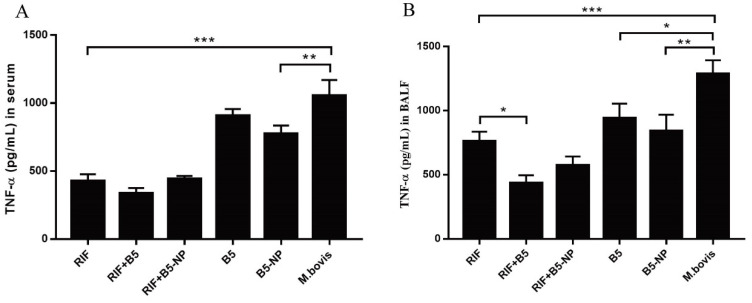
Effects of B5 combined with RIF on TNF-α secretion in serum and BALF. (**A**,**B**) Concentrations of TNF-α in serum (**A**) and BALF (**B**) were examined by ELISA. Serum and BALF were collected two weeks after the last treatment. All data showing represent the mean ± SD of three independent experiments (* *p* < 0.05; ** *p* < 0.01; *** *p* < 0.001).

**Table 1 pharmaceutics-12-01172-t001:** Characterization of PLGA-NPs with sizes and surface chemistries (n = 3).

PLGA-NPs	Size (nm)	PDI	Zeta Potential (mV)	EE (%)	LE (%)
PBS-NPs	193.2 ± 22.0	0.16 ± 0.03	−25.1 ± 1.5		
B5-NPs	206.6 ± 26.6	0.16 ± 0.04	−27.1 ± 1.5	85.5 ± 2.5	0.5 ± 0.02

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
