# Peer review of "Immunoregulatory and Antimicrobial Activity of Bovine Neutrophil β-Defensin-5-Loaded PLGA Nanoparticles against Mycobacterium bovis"

_pharmaceutics, 2020, doi:10.3390/pharmaceutics12121172_

Round 1
Reviewer 1 Report
The manuscript is well written, has a relevant analysis and contribution to the field, and should be of great interest to the readers.
The work is very complete and scientifically sound. The data is compelling and the overall work very detailed.
However, there are some small details that require further attention from the authors, namely the use of abbreviators and acronyms should be revised and defined correctly.
Reviewer 2 Report
This is a thorough study on β-defensin-5-loaded PLGA nanoparticles with interesting and important results. Before publication one important point has to be corrected or clarified:
Page 4, line 158: Authors write: ‘…totally 3 doses of B5 (50 μg in 50 μl of PBS, equivalent to 2.5 mg/kg ) or B5-NPs (1 mg B5-NPs in 50 μl of PBS, equivalent to 50 mg/kg )…’ B5 dose was 50 μg in 50 μl PBS, which means 2.5 mg/kg. The loading of B5-NP was 0.5 %, then 1 mg B5-NP should contain 5 μg B5. How can it be equivalent to 50 mg/kg (20 times higher than the B5)? Why was the dose of B5 in nanoparticles different from that of B5 in immunization study?
Minor comments:
Page 3, line 127: ’The size and zeta of NPs..’ should be corrected to ’ The size and zeta potential of NPs…’
Figures of Fig. 7. are too small and invisible.
Page 12, line 379 ‘Similarly, Morphological…’ should be corrected to ‘Similarly, morphological…’
Page 12, line 400: ‘…a growing number of studies provide…’ must be corrected to ‘…a growing number of studies provides…’
Reviewer 3 Report
The authors systematically conducted many experiments. Their effort deserves some credit. However, there are several concerns and deficiencies, as shown below.
Lines 14 – 19. There are too much general information belonging to introduction.
Lines 29 & 30, bacterial load in lung: We prefer to put ‘the’ in the front of an organ.
Lines 30 & 33. The short of rifampicin (RIF) is written first in line 30, which is followed by its full name and short in line 33.
Line 31, the number of bacterial in lung: Bacterial is adjective. The bacterial number in the lung.
Any information on the molecular weight of B5?
Lines 114 & 115: The molar ratio of L to G is mentioned twice.
2.3. Preparation of B5-NPs: One mg of B5 was dissolved in 0.5 ml of water, and the resultant solution was used as w1. After nanoparticle preparation, its encapsulation efficiency was determined to be 85.5 ± 2.5%. The theoretical load of B5 in PLGA NP is about 0.62% (100x1/161). Don’t the authors think this payload is too low? Also, isn’t there any problem in weighing 1 mg of B5 in terms of accuracy and precision?
Any concern on the propensity of B5 for being denatured upon exposure to the water/ethyl acetate interface and toward sonication?
2.4. Determination of B5-encapsulation and B5-loading efficiency: If we assume 100% protein encapsulation efficiency, 10 mg of B5-NP contain 6.2 mg of B5. Since B5 was dissolved in 50 ml of 1% SDS solution, its concentration is extremely low (0.12 mg/ml). What was the mixing volume ratio of the B5 sample solution to the micro-BCA working solution? What was the lowest and highest protein concentrations used for the authors’ standard calibration curve? Is it possible to provide the authors’ method validation data in the form of a supplementary material?
2.6. Cell culture and stimulation: The macrophages were stimulated for 24 hr with either B5 (5 mg/ml) or B5-NPs (100 mg/ml). However, the dose of B5 alone (5 mg) is not equivalent to that of B5-NPs (100 mg of B5-NPs contain 0.53 mg of B5). Furthermore, the in vitro study data indicates that about 45% of B5 is released out of B5-NPs within 24 hr (Figure 2 data). All these data suggest that, when the microphages were treated with B5 and B5-NPs, they were not simulated by the same amount of B5. Because of this consideration, the reviewer feels hard to agree to the authors’ result and discussion (lines 253 – 266 & Figure 3).
Lines 136 – 139 & line 238, Table 1: The authors use inconsistent abbreviated forms (EE, LE, EF, LC, EF, and LF). EE seems to be the same as EF, while LE/LC/LF are all the same. Please write consistent, appropriate short forms.
Line 224, EF and LE of 85.5 ± 2.5% and 0.5% ±0.02%. If average EE is 85.5%, average LE/LC should be 0.53.
Figure 2. What is the difference between D and E SEM micrographs? One has the bar size of 5 mm, and the other’s bar size is 500 nm.
Round 2
Reviewer 3 Report
All the concerns raised by the reviewer have been addressed accordingly.